# A Cross-Correlation Based Method for Determining Size-Resolved Particle Growth Rates

Janne Lampilahti[1], Pauli Paasonen[1], Santeri Tuovinen[1], Katrianne Lehtipalo[1,2], Veli-Matti Kerminen[1], Markku Kulmala[1]

[1]Institute for Atmospheric and Earth System Research, University of Helsinki, Helsinki, Finland
[2]Finnish Meteorological Institute, Helsinki, Finland

*Correspondence to:* Janne Lampilahti (janne.lampilahti@helsinki.fi)

**Abstract.** The particle growth rate (GR) is a key parameter in aerosol dynamics and plays a crucial role in understanding atmospheric new particle formation and its effects. A fast, robust and reproducible calculation of GRs from aerosol number-size distribution data remains a challenge. In this study, we introduce a new method that we call the maximum cross-correlation (MCC) method for calculating particle and ion GRs from number-size distributions. We employed this novel method to calculate GRs from simulated data and from ion and total particle number size distribution data from Hyytiälä, Finland. We compared our method against conventional methods for calculating the GRs and found good agreement. The MCC method enables fast and repeatable GR calculations from large aerosol datasets, which facilitates the systematic incorporation of GR analysis into new particle formation studies.

## 1 Introduction

New particle formation (NPF) is the process by which gas-phase molecules cluster and grow to form stable aerosol particles in the atmosphere (e.g. Kulmala et al., 2013). NPF plays a critical role in the climate system, as it serves as a major source of cloud condensation nuclei (CCN), which influence cloud properties and radiative forcing in the atmosphere (Gordon et al., 2017; Merikanto et al., 2009; Yu and Luo, 2009; Zhao et al., 2024).

Particle growth rate (GR), defined as the rate of change of particle diameter, is a key quantity characterizing NPF (Kulmala et al., 2012). GR is an important parameter when determining the probability that the freshly formed particles reach CCN sizes, especially when the particles are only a few nanometers in diameter and easily scavenged by the pre-existing aerosol (e.g. Cai et al., 2022; Kerminen et al., 2018; Stolzenburg et al., 2018, 2023; Yli-Juuti et al., 2011).

In ambient observations the GR is often determined from particle or ion concentration measurements that are resolved by both size and time. A spatial homogeneity assumption is often made,

which states that the determined GR is equal to the average GR of the aerosol particle population. The
methods to calculate GR can be roughly divided into collective and dynamic methods (Stolzenburg et
al., 2023).
In the dynamic approach one predicts the evolution of the aerosol size distribution using
general dynamic equation (GDE) and compares the result to the observation (e.g. Pichelstorfer et al.,
2018). The difference between the GDE prediction and observation is attributed to condensational
growth as other processes causing apparent growth are included in the GDE. The dynamic approach
can in principle be fully automated. However, the processes included in the GDE can be difficult to
quantify in the atmosphere and the solutions to GDE are highly sensitive to measurement uncertainty,
which poses issues especially in sub-5 nm sizes.
The collective methods can be further divided into representative diameter and representative
time based methods. Usually a line is fitted through the representative time-diameter pairs in a selected
size range and the corresponding GR is reported as the slope of the fit. In the collective methods one
determines a so-called collective or apparent GR, which represents the observed growth of the particle
population at the measurement coordinates over the duration of the NPF event. The apparent GR does
not separate out any processes responsible for the observed particle growth (e.g. condensational growth,
coagulation or size-dependent loss processes).
In representative diameter based methods the GR is determined by analyzing the number size
distribution across time and finding representative diameters that are then used at calculating the GR.
In the so-called mode fitting (MF) method (Dal Maso et al., 2005; Kulmala et al., 2012) a log-normal
distribution is fitted to the growing particle mode at successive time steps and the rate of change of the
peak values is used at calculating the GR. The representative diameter based methods work best when
the growing particle mode is fully visible in the data. In sub-5 nm sizes this is often not the case and
the MF method will underestimate the GR. Paasonen et al. (2018) developed an automatic method
based on mode fitting that identifies growing particle modes and calculates their GRs. However, the
method is less reliable for determining GRs in the smallest particle sizes at the onset of NPF events.
In representative time based methods, the GR is determined by analysing the concentration
time series across different size channels and finding representative times, which are then used in the
GR calculation. A conventional approach is to fit a function to the increasing concentrations associated
with the growing particle mode and use specific features from the fitted curves as representative times.
In the maximum concentration (MC) method, a Gaussian function is used, and the peak positions
determine the GR (Hirsikko et al., 2005). In the appearance time (AT) method, a sigmoid function is
fitted to the leading edges of the rising concentrations, and the midpoint values are used to calculate
the GR (Lehtipalo et al., 2014). The AT method is suitable also for cases when NPF is sustained over
longer time periods, e.g. during chamber experiments (Dada et al., 2020), and therefore Gaussian
function is not suitable for locating the maximum concentration.
A similar approach to the representative time based methods is to estimate the representative
time lag between the rising concentrations in two separate size channels (Riccobono et al., 2012; Sihto

et al., 2006). The GR is calculated by dividing the size difference with the time lag. The difference between finding representative time lags and representative times is subtle but relevant since in practice the methodology can be different and one may be more amenable to automation than the other. The representative time (lag) based methods are suitable for instruments with limited number of size-channels and when the particle mode is not fully visible in the data, which is usually the case when the growing particles are sub-5 nm in size. Representative time (lag) based methods do not require the absolute concentrations to be correct and can even work on raw signals.

In both representative diameter and representative time based methods automating the fitting procedure for NPF events is challenging. The range of data around the NPF event used for the fitting is usually manually selected by the researcher. This makes GR calculation labor-intensive and subjective. Fitting-based methods can be very sensitive to the chosen data range and are influenced by for example concentration spikes from local emissions, instrumental noise and meteorological conditions. All this leads to variability in GR estimates and reduced reproducibility. In part due to these limitations, despite the abundance of aerosol number-size distribution data, comprehensive datasets that report GRs in different size ranges are scarce and the methods used can be poorly documented.

In this study we introduce the maximum cross-correlation (MCC) method, which is an automatic representative time lag based method for GR calculation. Our objective is to validate the new method using simulated data and by comparing it against conventional methods.

## 2 Methods

### 2.1 Maximum cross-correlation method

When no other processes significantly influence the particle size distribution, particle growth during or after an NPF event leads to an increase in particle number concentration that is observed earlier in the smaller size channels and later in the larger size channels. The task is to find a way to calculate the time lag between the concentration rise for the different size channels and use it in the calculation of the GR.

We assume that a good condition for finding the representative time lag is when the concentrations in two size channels are maximally correlated. Next we will outline how this idea is used to calculate particle GRs from number-size distribution data.

1. Let us choose particle diameters $d_1$ and $d_2$ ($d_1 < d_2$) from the number size distribution
2. From the number-size distribution interpolate the concentration time series $N_1(t)$ and $N_2(t)$ that correspond to the size channels represented by the chosen diameters.
   a. It is possible to select an arbitrary time window for the time $t$. The time window can be different for the small and large diameters. Here we chose one day from midnight to midnight local time since NPF tends to follow a diurnal cycle in most environments.

b.   Here we required that no more than 5% of the concentration values along each size

channel in the size range of interest were missing, otherwise the day was categorized

as bad data. Days when the instrument was not measuring were also categorized as

bad data.

3.  Calculate normalized cross-correlation for $N_1(t)$ and $N_2(t)$:

$$R_{N_1 N_2}(\tau) = \frac{1}{M(\tau)^{\gamma}} \sum_{t} \frac{(N_1(t-\tau)-\mu_1)(N_2(t)-\mu_2)}{\sigma_1 \sigma_2}$$

a.   $N_1(t)$ and $N_2(t)$ are normalized by subtracting their means $\mu_1$ and $\mu_2$ and dividing

by their standard deviations $\sigma_1$ and $\sigma_2$. This makes the method less sensitive to

baseline concentration levels or the differences in concentration amplitude.

b.   $\tau$ is the time lag and $\tau_{lim}$ is the maximum allowed lag. We varied $\tau$ at increments of

$\Delta\tau = 1$s.

c.   $M(\tau)$ is the number of overlapping data points between time series $N_1(t)$ and $N_2(t)$

for a given $\tau$. The summation is only over the overlapping data points so a higher

degree of overlap will lead to a higher cross-correlation. In order correct for this

effect we divide the sum (raw cross-correlation) by $M(\tau)^{\gamma}$ (overlap correction). $\gamma$ is

an exponent that determines how aggressive the overlap correction is. An optimal

value for $\gamma$ can be determined from the data (see Section 3.1).

i.   $\gamma > 1$ overlap correction is amplified as $\tau$ increases.

ii.   $\gamma = 1$ overlap correction is directly proportional to $\tau$.

iii.   $0 < \gamma < 1$ overlap correction is dampened as $\tau$ increases

iv.   $\gamma = 0$ same as raw cross-correlation

130        d.   Depending on the data quality, the channel concentrations $N_1(t)$ and $N_2(t)$ may have

to be smoothed for example by using a rolling mean.

4.  Find the representative time lag as the time lag at maximum correlation $\tau_{max} =$

$arg\ max(R_{N_1 N_2}(\tau))$

a.   Return a missing value for results where $\tau_{max} \leq 0$s and for the limiting case $\tau_{max} =$

$\tau_{lim}$

5.  Calculate the growth rate as $GR_{d_1-d_2} = \frac{d_2 - d_1}{\tau_{max}}$

Figure 1 illustrates how the MCC method is used to calculate particle GR from the number

size distribution. In this case we calculated the GR of negative ions from 2 nm to 3 nm. The MC method
was also applied to the case for comparison.

If the diameters $d_1$ and $d_2$ are too widely spaced, the correlation between the size channels is

influenced by unrelated atmospheric processes. This is why the size range may have to be divided into
smaller size ranges or bins and then the MCC method can be applied to each smaller bin separately. In
order to calculate GR for the whole size range one should add the representative time lags $\tau_{max,i}$ for
each bin numbered by $i$

$$\tau_{max} = \sum_{i=1}^{n} \tau_{max,i}$$

and use the resulting $\tau_{max}$ in the final GR calculation. Here we used a condition that if for any of the
bins $\tau_{max,i} \leq 0$ or $\tau_{max,i} = \tau_{lim}$ then the GR for the whole size range would also be a missing value.

The number of bins should not exceed the number of size channels available in the data and

the concentrations at bin edges can be found by linear interpolation. Each bin should be of equal width
on log-scale.

## 2.2 Hyytiälä dataset

The MCC method can be used on individual days when a growing particle mode is present or to
determine the distribution of GRs across multiple days in an automated fashion, as demonstrated in this
study.

We tested the method on an ion and total particle number size distribution dataset from the

SMEAR II station. SMEAR II station is located in Hyytiälä, Finland in a rural boreal forest
environment (24.30E, 61.85N, 180m; Hari and Kulmala, 2005). The dataset is approximately 14 years
long (Feb 2010-Dec 2024). No NPF event classification was done on the days in the dataset prior to
the GR analysis. The ion and particle number size distributions were measured by a Neutral cluster and
Air Ion Spectrometer (NAIS, Ariel Ltd.; Mirme and Mirme, 2013) and Differential Mobility Particle
Sizer (DMPS; Aalto et al., 2001).

The NAIS measured the number size distribution of air ions and total particles in the mobility

equivalent diameter range of approximately 0.8-40 nm and 2.5-40 nm respectively. The DMPS system
measured the number size distribution of total particles in the mobility equivalent diameter range of 3-
1000 nm. The number size distributions were averaged to 1 hour time resolution.

GRs were calculated in three size ranges: 2–3 nm, 3–7 nm and 7–20 nm. Each size range was

separately subdivided into smaller bins on log-scale: two bins for 2–3 nm, three for 3–7 nm, and four
for 7–20 nm. In the NAIS data there were 6, 12 and 15 size channels between 2-3 nm, 3-7 nm and 7-
20 nm. In the DMPS data there were 7 and 9 size channels between 3-7 nm and 7-20 nm. The
concentrations at bin edges were interpolated from the number size distribution (number concentrations
normalized by logarithm of size channel width, $dN/d\ log_{10} d$) measured by the NAIS or the DMPS.

We compared our results against a pre-existing GR dataset from Hyytiälä. In this dataset GRs

were calculated using the MC method from NAIS ions (2003-2019) in the 1.5-3 nm, 3-7 nm and 7-20
nm size ranges and from DMPS total particles (1996-2024) in the 3-7 nm and 7-25 nm size ranges. The
overlapping time range was Jan 2010-Aug 2019. In addition GR datasets mainly reporting descriptive
statistics have been published from Hyytiälä (Gonzalez Carracedo et al., 2022; Hirsikko et al., 2005;
Manninen et al., 2009, 2010; Yli-Juuti et al., 2011), which serve as another point of comparison.

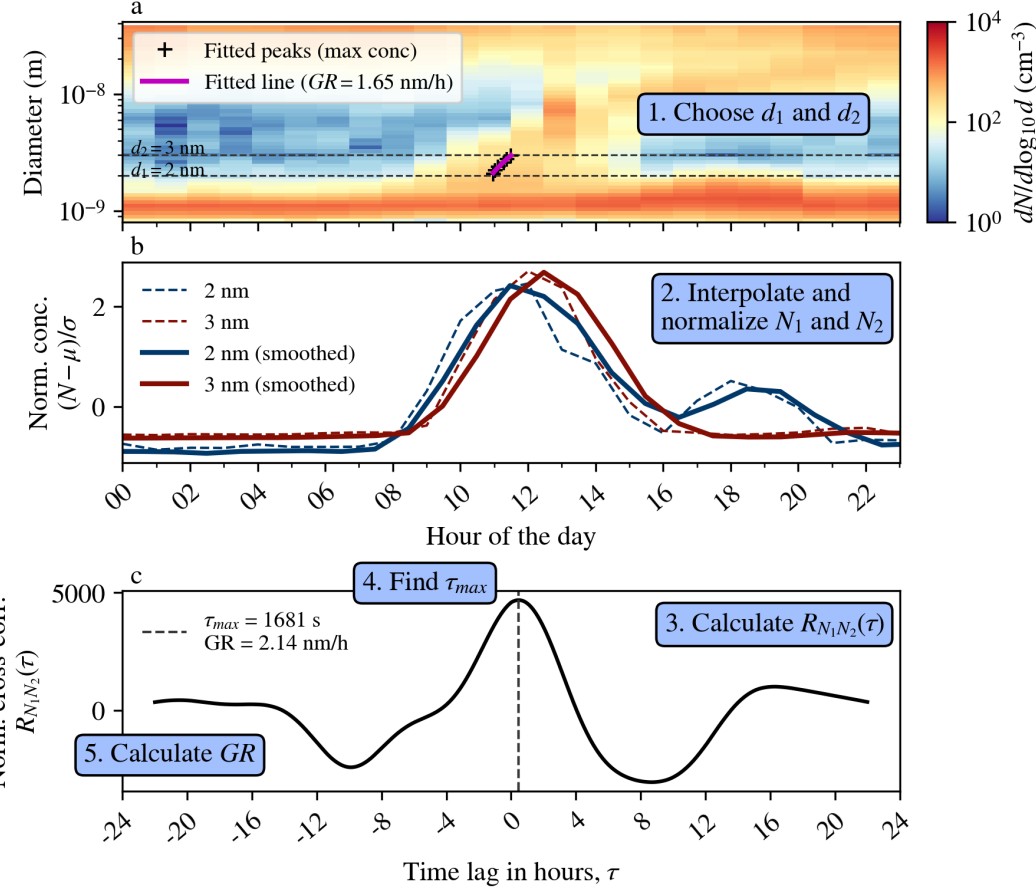

**Figure 1: Example case from 15 March 2010 illustrating the calculation of GR for 2-3 nm negative ions using the MCC method. Panel a shows negative ion number size distribution measured by the NAIS with peak diameters fitted using the MC method and the GR found by fitting a line to them. Panel b shows normalized particle concentrations in the two size channels with and without a 3 hour rolling mean applied. Panel c shows normalized cross-correlation between size-channels, with the $\tau_{max}$ and corresponding GR shown. For normalization a gamma value of 0.25 was used. The size range was not divided into smaller bins.**

## 2.3 Synthetic dataset

The ARCA box model (Atmospherically Relevant Chemistry and Aerosol model; Clusius et al., 2022) was applied to simulate a NPF event with a known condensational GR, which served as a reference for evaluating different GR calculation methods. The model configuration file can be found in the supplementary material.

## 3 Results and discussion

### 3.1 Sensitivity to input parameters

Dividing the raw cross-correlation by $M(\tau)^\gamma$ may introduce a bias towards larger or smaller GRs depending on the value of $\gamma$. In order to study this effect and find an optimum value for $\gamma$ we investigated if there was a systematic shift in the median of the GR distribution for different values of $\gamma$ when the size range was divided into more bins (Figure 2a). For comparison we also tested if the GRs calculated from the simulated NPF event would shift under similar conditions (Figure 2b).

Both the GR distribution medians and the simulated GRs showed similar shifts for different $\gamma$ values. In both cases the value that had minimal effect on the GR was $\gamma = 0.25$, which is the value we used in the rest of the analysis.

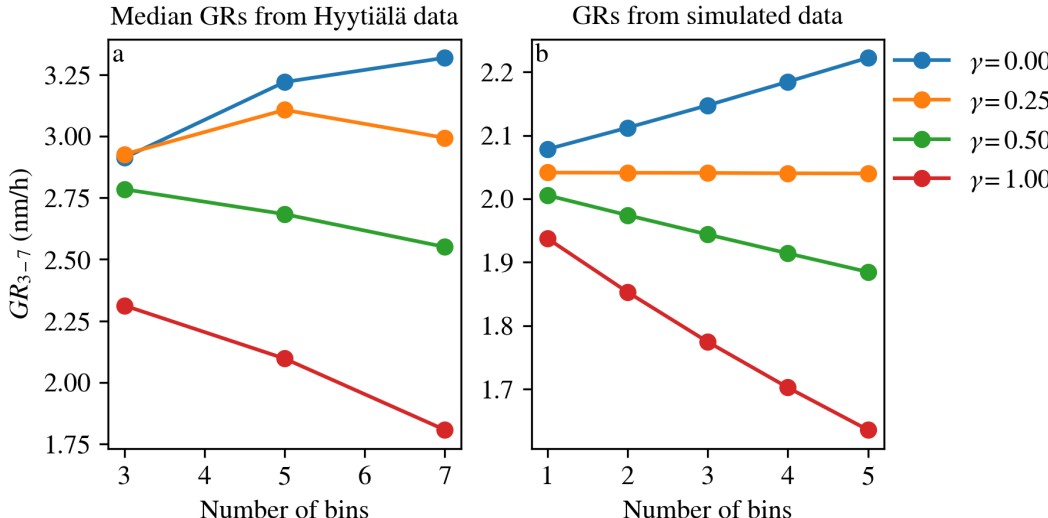

**Figure 2: Panel a shows the median GRs (3-7 nm) from Hyytiälä negative ion data for different values of $\gamma$ as a function of the number of bins in the size range. Panel b shows the GRs (3-7 nm) calculated from the simulated NPF event for different values of $\gamma$ as a function of the number of bins in the size range.**

Figure 3 shows the effect of the number of bins ($n$), maximum allowed lag ($\tau_{lim}$) and smoothing window width ($S$) on the resulting GR distributions.

Figure 3a shows that a higher number of bins increases the signal-to-noise ratio but reduces the number of days in the GR distribution. This is because if there are more smaller size ranges the probability of getting $\tau_{max,i} = 0$ s (or other invalid value) is increased and in this case a missing value is returned for the whole GR. In order to maximize the signal-to-noise ratio and minimize the number of discarded GR days, we aimed for a bin width of approximately 0.1 on the log scale. For 2-3 nm, 3-7 nm and 7-20 nm this corresponded to $n = 2$, $n = 3$ and $n = 4$.

Figure 3b shows that if the range of allowed lags is too narrow some low GR values are cut out. If the lag range is increased enough to capture all the GRs, then further increases in the maximum

allowed lag have no effect on the shape of the GR distribution. In the analysis we used $\tau_{lim} = 22$ h, which should be enough to capture all particle growth during the day.

Figure 3c shows that increased smoothing window width shifts the GR distribution towards lower GR values. This is a known effect in the AT method as well (Lehtipalo et al., 2014). It may be that increased noise shifts the GR distribution towards higher GR values. In the analysis we used $S = 3$ h, which should be enough to remove some of the random concentration peaks and noise while preserving the gradual concentration increase due to NPF.

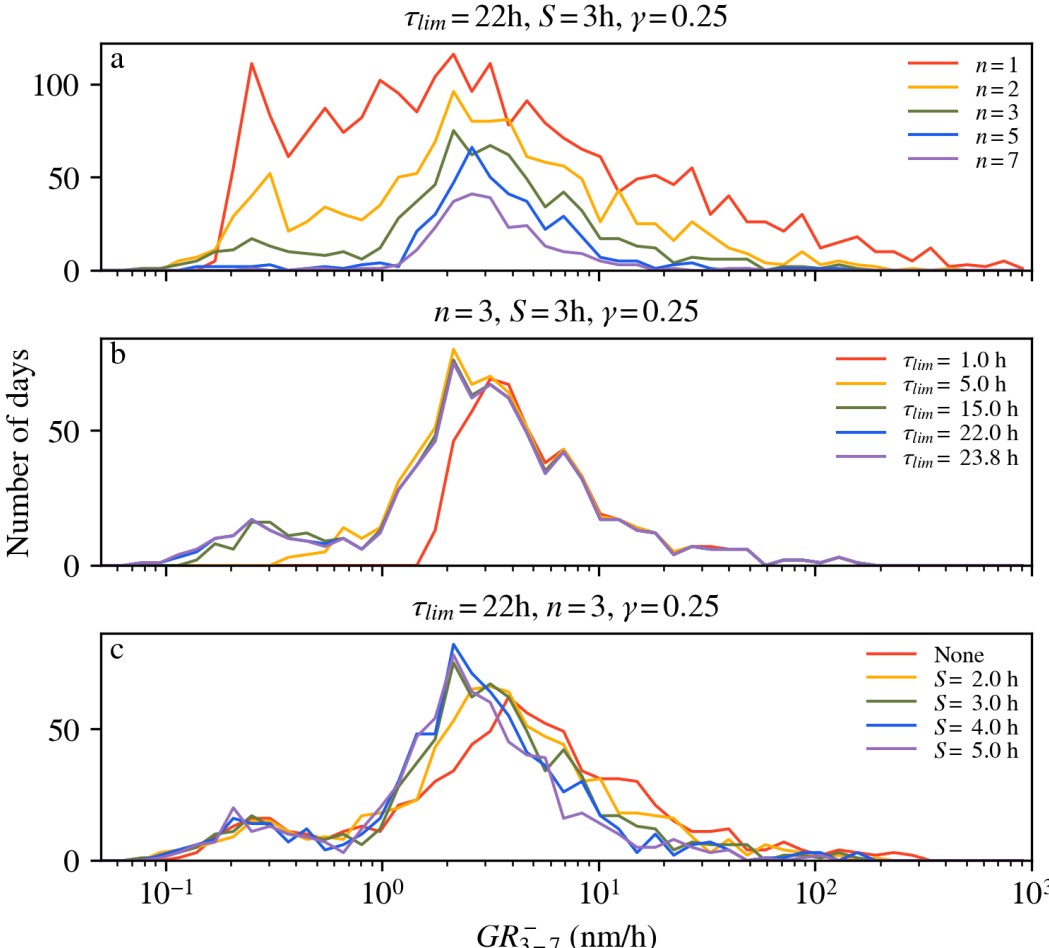

Figure 3: Panels a, b and c show the effect of number of bins ($n$), maximum allowed lag ($\tau_{lim}$) and smoothing window width ($S$) on the GR distributions of 3-7 nm negative ions respectively. The input parameters that were kept constant are shown on top of each panel.

## 3.2 Simulated NPF event

The simulated NPF event is shown in Figure 4. Figure 4a shows the evolution of the number-size distribution along with representative time-diameter pairs fitted using the MC, AT and MF methods.

Figure 4b shows the mean (condensational) GR in each size bin from the model and the GRs obtained
using MCC, MC, AT and MF methods.

All methods underestimated the true GR. The MCC, MC and AT methods showed less

underestimation and were in good agreement with each other. As expected, the MF method
underestimated substantially more below 5 nm.

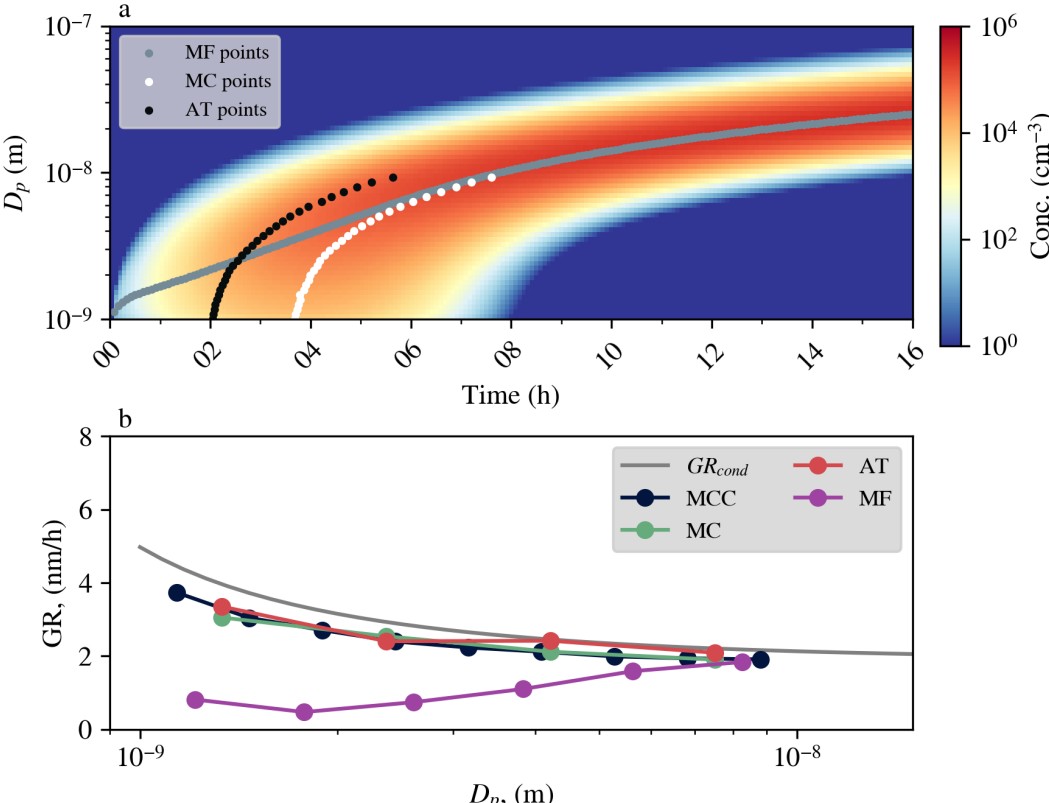


**Figure 4: The NPF event simulated using the ARCA box model. Panel a shows the time evolution of the**

**number-size distribution and the representative time-diameter pairs fitted using MC, AT and MF methods.**

**Panel b shows the comparison of different GR methods against the true (condensational) GR obtained from**

**the model.**


### 3.3 Hyytiälä GR distributions

The GR distributions obtained from the 14-year dataset in Hyytiälä using the MCC method are shown
in Figure 5. The $GR_{2-3}$ and the $GR_{3-7}$ distributions in both ions and total particles appear to consist
of two distinct parts that we call the "background" and the "signal". Background refers to the smaller
peak at lower GR values and signal to the larger peak at higher GR values.

We assumed that the signal peak contains days with true particle growth and separated it from

the background by identifying a local minimum between them. There can be several reasons for the
background distribution at very low GRs. For example even with the optimized $\gamma$ exponent, there could
remain some bias towards high time lags. In the absence of NPF events the concentrations in the small
and large size channels follow a diurnal cycle based on the boundary layer (BL) height with higher
concentrations in the early morning and late evening when the BL is low and lower concentration
during midday when the BL is high. Cross-correlation between the early morning and late evening can
therefore be dominating when there is no NPF leading to a distribution of low GR values.

To illustrate this further, Figure 5 shows the GR distributions on days with low (bottom 80%)

and high (top 20%) nanoranking values (Aliaga et al., 2023). On days with high nanorank it is more
likely that intense NPF takes place. The GR distributions on high nanorank days clearly have a reduced
background component compared to the GR distributions on low nanorank days. Due to the
concentration normalization MCC method can return GRs also on so-called quiet NPF event days
(Kulmala et al., 2022), which can have low nanorank values.

We also prepared Figures S1-S4 (supplementary material), which show examples of negative

ion number size distributions on days that are from the signal distribution and on days that are from the
background distribution for 2-3 nm and 3-7 nm size ranges. The majority of the signal days, especially
looking at the larger size range, show features typical of NPF event days, while the background days
contain mostly days that would be classified as nonevent or undefined days (Dal Maso et al., 2005).
This supports excluding them from the GR analysis.

The number of large outliers was relatively small (<20) in all GR distributions, which suggests

that most local particle plumes that might cause very high GRs are filtered away by the requirement
that $\tau_{max} > 0$ s. Averaging the data to 1 hour time resolution probably also reduced the number of
large outliers in the GR distribution. We chose to ignore the large outliers due to their small number.

The median GRs in Figure 5 showed an increasing trend with particle size, a pattern

commonly observed across various environments (Kerminen et al., 2018; Stolzenburg et al., 2023). In
the same size range the median GRs found for the positive ions, negative ions and total particles (from
DMPS) were similar, which is also supported by previous observations (Gonzalez Carracedo et al.,
2022; Hirsikko et al., 2005; Manninen et al., 2009; Yli-Juuti et al., 2011). The GRs calculated from the
NAIS total particles were higher compared to the other GRs in the same size range. This might be
linked to the NAIS measuring higher concentrations in total particle mode compared to other
instruments (Kangasluoma et al., 2020).

Further testing needs to be done by using data from environments that have extremely slow

(e.g. Arctic sites) or fast (e.g. some coastal sites) particle growth to see if a similar separation into
background and growth distributions occurs. In urban environments the number of outliers may be
higher due to increased local particle emissions.

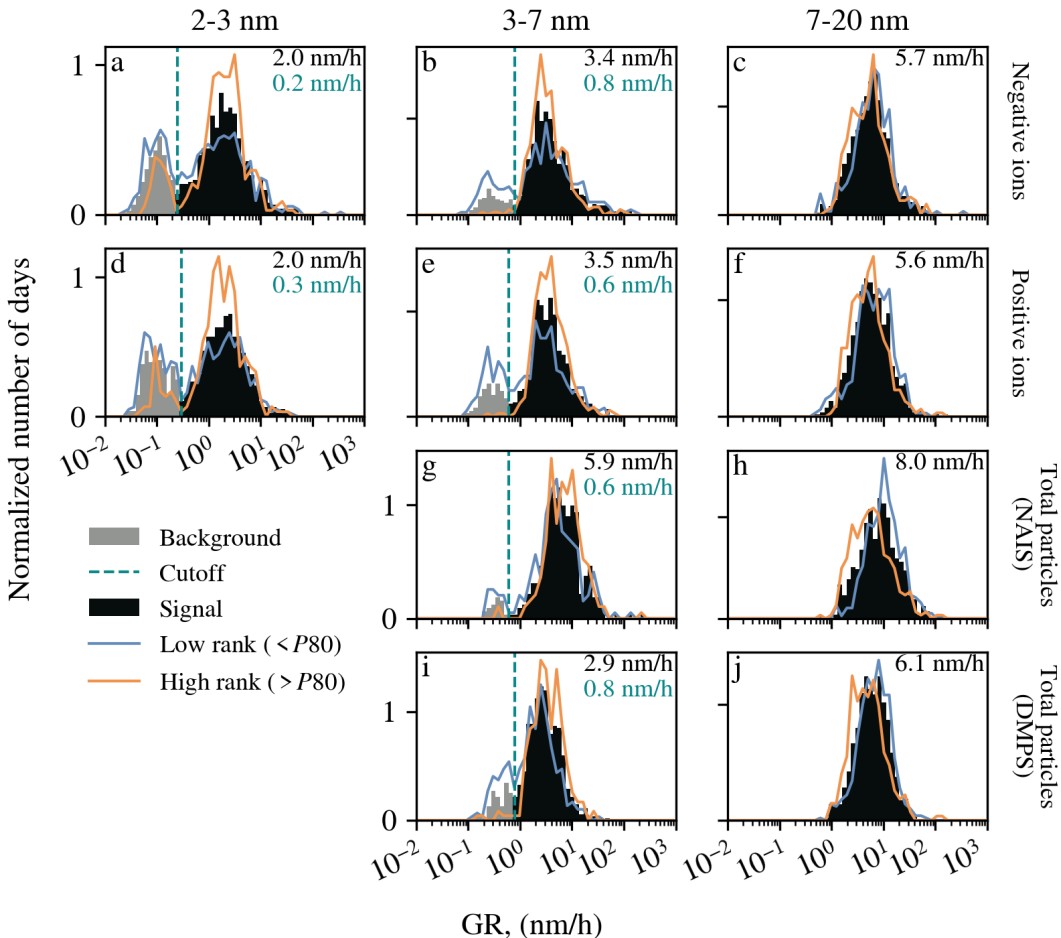

**Figure 5: The GR distributions from Hyytiälä, separated into background and signal parts at the cutoff (dashed line). The median GR from the signal distribution is shown in the top right corner. The blue and orange lines show the GR distributions on days that have low (<P80) and high (>P80) nanorank respectively. For comparability all distributions were normalized to unity.**

Figure 6 shows that there were many days when GR was not calculated due to too much missing data (bad data) or due to the method returning a missing value due to $\tau_{max} \leq 0$ s (negative or infinite GR) or $\tau_{max} = \tau_{lim}$ (limiting case). The relatively large number of bad data is explained by our rather strict criteria for usable data (<5% missing data in the size range of interest). Days when the instrument was not measuring were included in the bad data.

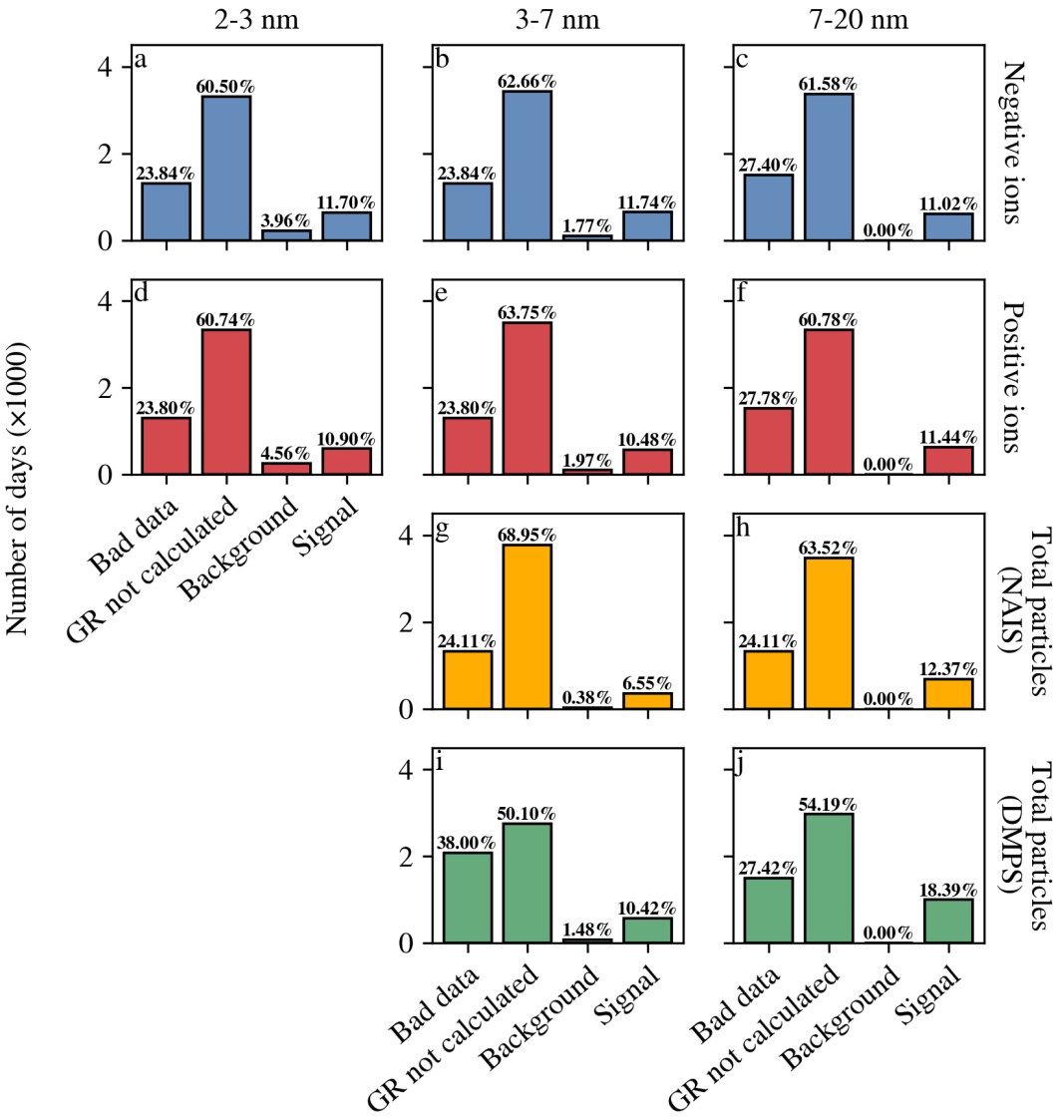

**Figure 6: The number of days categorized as bad data, GR not calculated, background and signal. Percentage of total days is shown on top of each bar.**

## 3.4 Comparison against previous results

Figure 7 shows a comparison between the MCC results and the MC results from Hyytiälä. Each point corresponds to a day when both methods successfully returned a GR in the given size range. GRs from both methods cluster around the 1-to-1 line and show a clear positive correlation.

In the smallest size range the number of common data points is reduced and some more outliers are present. This is likely because reliably obtaining GRs is more difficult in sub-3 nm sizes due to the increased measurement uncertainty.

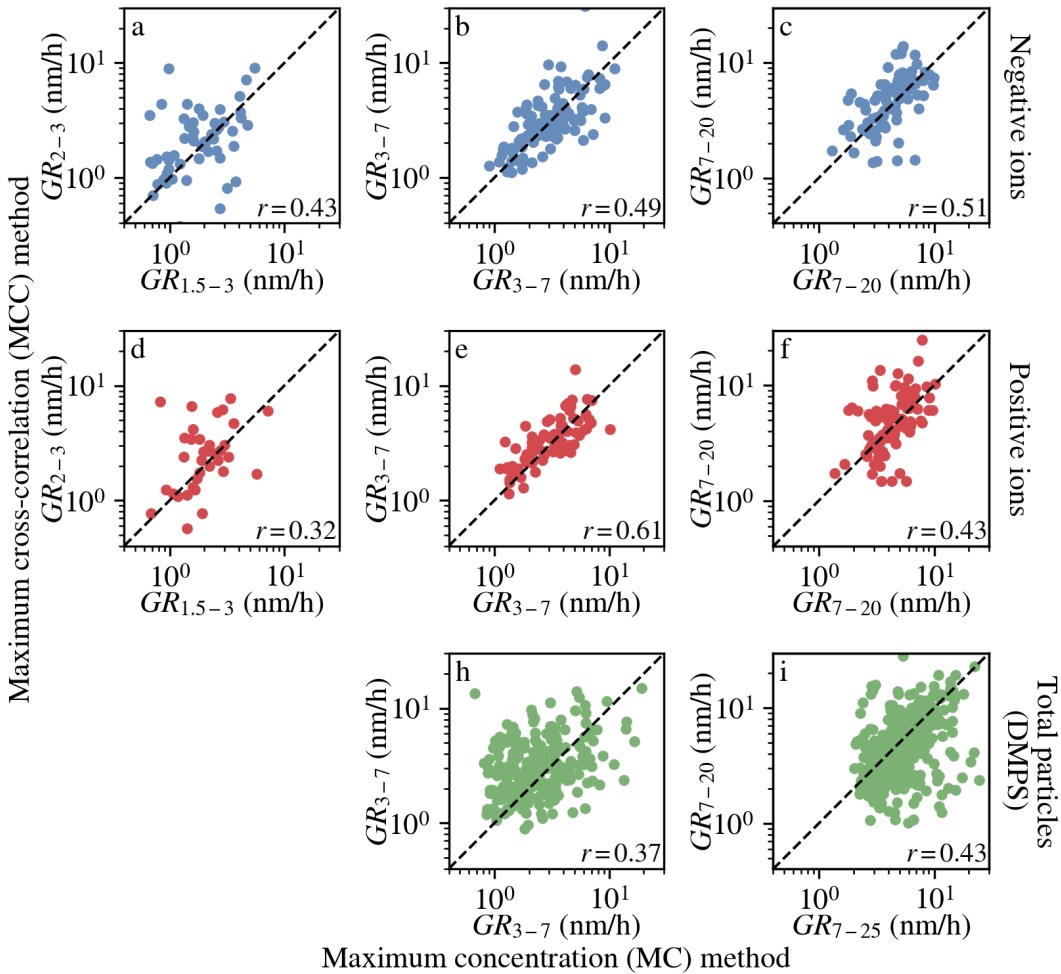

**Figure 7: Correlation between ion and total particle GRs obtained from the MCC and MC methods. Each point represents a day when both methods returned a valid GR in the given size range. The GRs were calculated from Hyytiälä data between 2010-2019. The Pearson correlation coefficients are shown in the lower right corner of each subplot.**

In Figure 8 we compared the percentages of days when only the MC method, only the MCC method or both methods returned a GR in the different size ranges for the time period 2010-2019. In each case the MCC method found significantly more GRs than the MC method. For the two larger size ranges, the MCC method identified most of the GR days detected by the MC method (73–87%). For the 3–7 nm total particle case, the MCC method captured only about 56% of the GR days detected by the MC method.

In the smallest, sub-3 nm, size range the MCC method found about half of the GR days found by the MC method. This is likely because the MC method was only applied to NPF events where the ion growth continued to larger sizes whereas the MCC method potentially also found cases of highly localized ion growth where the growth to larger sizes was not observed (e.g. Rose et al., 2018; Tuovinen et al., 2024).

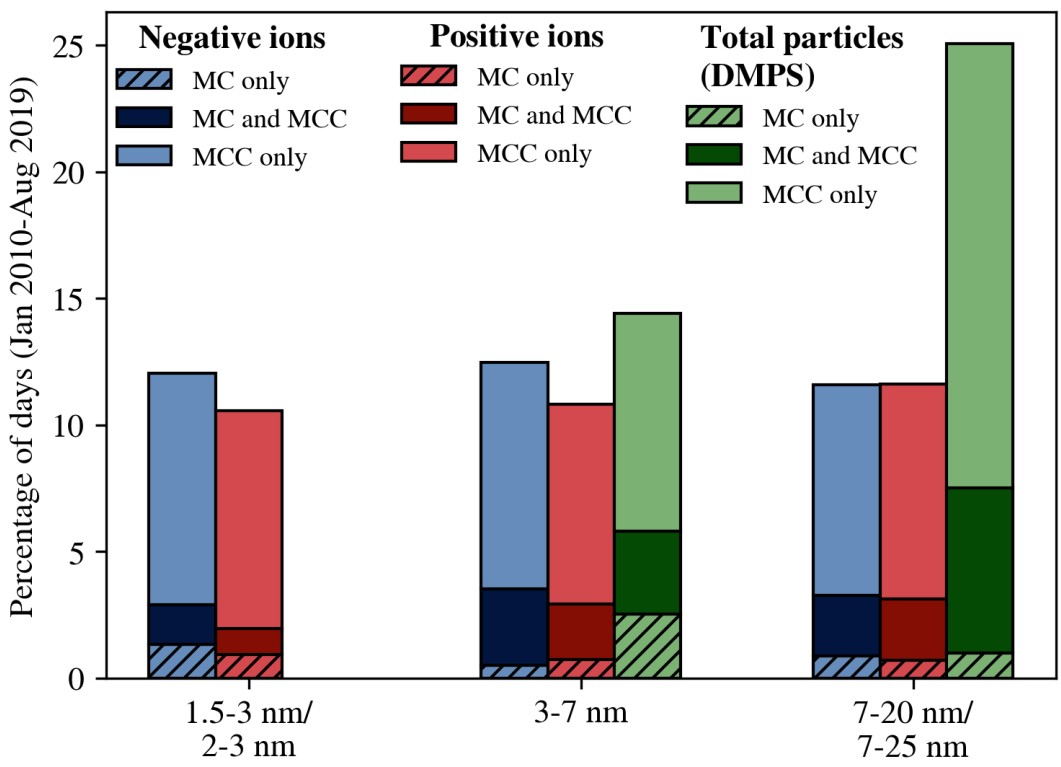

**Figure 8: Percentage of days during 2010-2019 when only the MC method, only the MCC method or both**
**returned a GR in a given size range. The bad data days were removed from the comparison.**

In Figure 9 we investigated how the days when the MCC method returned a valid GR or a missing
value were distributed in terms of the $\Delta N_{2.5-5}$ metric used in the nanoranking analysis (Aliaga et al.,
2023) and in terms of the traditional NPF event classification scheme (Dal Maso et al., 2005).
Figure 8a shows that GR days had higher $\Delta N_{2.5-5}$ metrics compared to the days with missing
GRs, indicating that there was more intense NPF on the GR days. This is supported by Figure 8b, which
shows that most GR days in the 3-7 nm and 7-20 nm size ranges were also NPF event days whereas
days with a missing GR value were more likely to be nonevents. In the undefined category the GR days
and the missing GR days were more evenly distributed.
Figure 8b shows a less clear trend in the 2-3 nm size range across the NPF event classes. This
can be because not all sub-3 nm ion growth continues to larger sizes and therefore will not be classified
as a NPF event in the classification scheme. The NPF event classification is done based on DMPS data
using total particles larger than 3 nm.

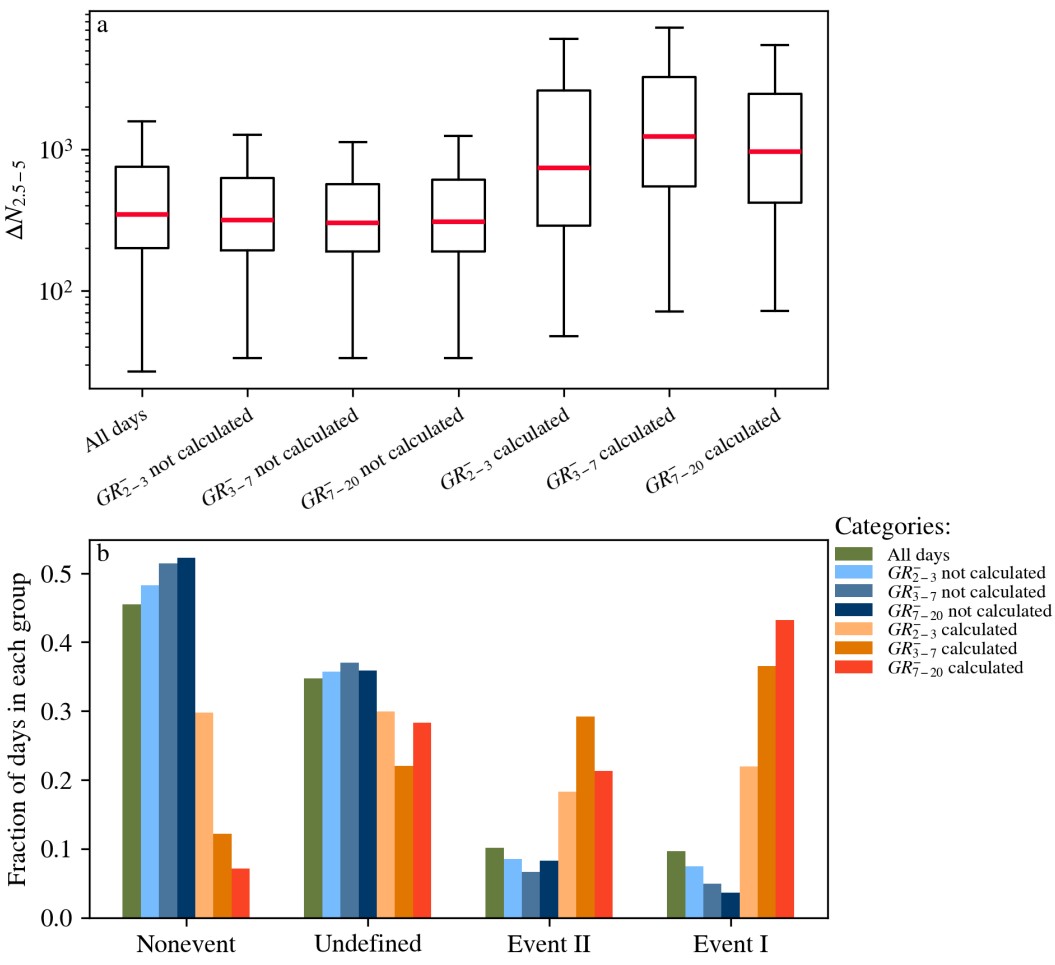

**Figure 9: Panel a shows the box plots of $\Delta N_{2.5-5}$ metrics from the nanoranking analysis in different categories. Panel b shows how the days in the same categories were distributed in terms of the NPF event classes. The bad data days were removed from the comparison.**

Finally, we compared our results with published GR data from Hyytiälä. Yli-Juuti et al. (2011) investigated ion and total particle GRs during NPF events in Hyytiälä (2003–2009) using data from similar instruments as in this study. GRs were derived with both the MC and MF methods, which showed good agreement, and the reported median GRs represent averages across instruments and methods. Ion data was used for the sub-3 nm sizes and both ion and total particle data for larger sizes.

Hirsikko et al. (2005), Manninen et al. (2009) and (2010) used similar methodology (only MC method) to investigate GRs in Hyytiälä between 2003-2004, March-June 2007 and March 2008-May 2009 respectively.

Gonzalez Carracedo et al. (2022) used the AT method and the MC method to calculate GRs in two different size ranges from ions measured by NAIS and total particles measured by DMA train. The authors found good agreement between the two methods. The data was measured in Hyytiälä between March-September 2020. For comparison purposes we chose only the ion GRs to calculate the

median GR in the smallest size range (1.8-3.2 nm) and both ion and total particle GRs to calculate the median GR in the larger size range (3.2-8 nm).

From our results we calculated the median 2-3 nm GR from ions and the median 3-7 nm and 7-20 nm GRs using ion and DMPS data. We used the 0 and 99 percentile values as our min and max values. The results of the comparison are shown in Figure 10.

Overall the median GRs in all the size ranges compare well across all studies. Also the min-max ranges and the interquartile ranges of the GRs (if reported) compare well across the studies. In the previously published studies the GRs were determined only on days with intense NPF, whereas our method is in principle also capable of finding GRs on days with less intense NPF. In Hyytiälä the GR is expected to be similar despite the intensity of NPF (Kulmala et al., 2022).

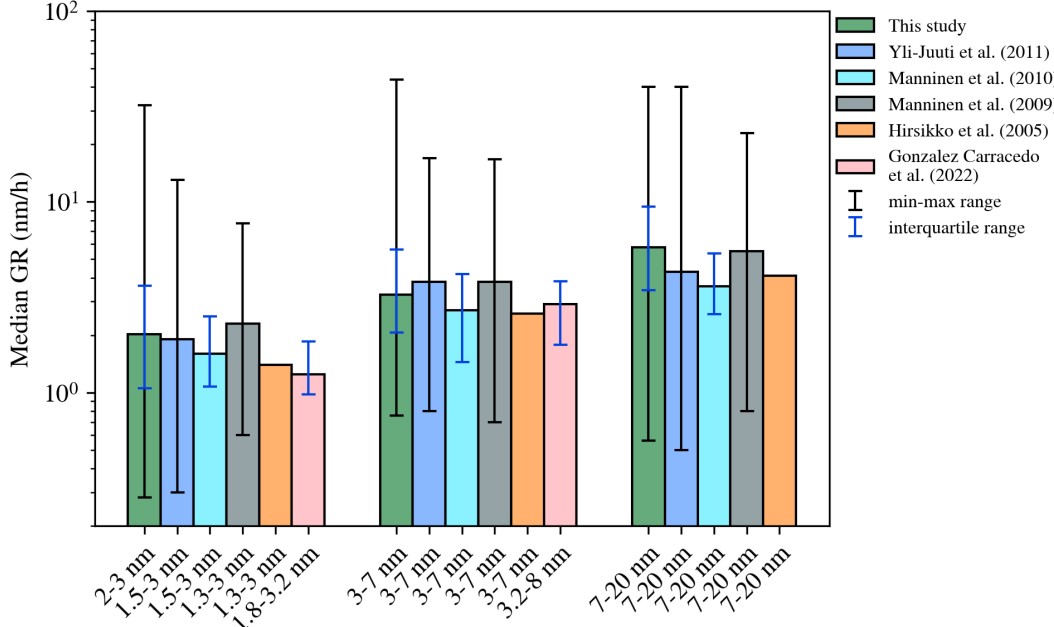

**Figure 10: Comparison of GRs obtained in this study against published data from Hyytiälä. The bar height is the median GR in each size range and the error bars show different ranges of GR values reported in the studies.**

**4 Conclusions**

We presented the MCC method for determining size-resolved particle GRs from aerosol number-size distribution data. The method was tested and validated on sub-20 nm ion and total particle number-size distribution data measured from a Finnish boreal forest. The proposed method uses cross-correlation to determine the representative time lag used for the GR calculation. We applied the method to a simulated NPF event with a known condensational GR and to approximately 14 years of ion and total particle data from the SMEAR II station in Hyytiälä, Finland. We calculated GRs in three different size ranges: 2-3 nm, 3-7 nm and 7-20 nm.

The GRs obtained using the MCC method compared well against the GRs obtained using the conventional methods in both the simulated data and in the measured data. Most days when the MCC method returned a GR were NPF event days in Hyytiälä.

One should keep in mind that the MCC method gives the apparent GR, which is roughly equal to the condensational GR in non-polluted environments with roughly homogenous sources of condensable vapors in surrounding areas. In polluted environments the effect of coagulation on the GR should be taken into account (Cai et al., 2021). Also heterogeneities in condensable vapor concentrations upwind the observation site can have strong effects on the apparent GR, which can influence the interpretation of the results (Hakala et al., 2023; Kivekäs et al., 2016).

In the future it is important to test the MCC method using aerosol data from other types of environments. For example in places with local emission sources there may be multiple particle growth events during a single day. In this case running the MCC method using different time windows may be necessary in order to separate out the different growth processes. On the other hand, the suitability for detecting very low or very high GRs; as well as weak NPF should be further investigated.

The MCC method allows one to efficiently and systematically calculate GRs from ion and total particle number size distributions. The MCC method is readily applied to large collections of data, which facilitates the GR analysis from new and existing aerosol datasets. When combined with statistical NPF classification methods, such as the nanoparticle ranking method (Aliaga et al., 2023), it could replace the conventional labor-intensive NPF analysis, especially when NPF is the dominant source of particles in the size range.

**Data availability.** The NAIS dataset is available at https://doi.org/10.5281/zenodo.15648699 (Lampilahti et al., 2025). The DMPS dataset can be accessed through the SmartSMEAR data portal at https://smear.avaa.csc.fi/.

**Author Contribution.** JL conceived the method and conducted the data analysis. ST ran the simulation. PP and MK contributed to refining the methodology. JL wrote the manuscript with input from all co-authors.

**Competing interests.** Some authors are members of the editorial board of Aerosol Research. The authors have no other competing interests to declare.

**Acknowledgements.** We acknowledge the University of Helsinki support via ACTRIS-HY. We acknowledge the staff at the SMEAR II field station for their valuable support in maintaining the measurement infrastructure and assisting with data collection.

**Financial support.** ACCC Flagship funded by the Academy of Finland grant number 337549 (UH), "Gigacity" project funded by Wihuri foundation, European Research Council (ERC) project ATM-

GTP Contract No. 742206, Research Council of Finland University Profiling funding InterEarth (grant
no. 353218), Research council of Finland project CO-ENHANCIN (project number 360114) and
Horizon Europe project FOCI (project number 101056783).

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

Methods for determining particle size distribution and growth rates between 1 and 3 nm using the
Particle Size Magnifier, Boreal Environ. Res., 19, 22, 2014.
Manninen, H. E., Nieminen, T., Riipinen, I., Yli-Juuti, T., Gagné, S., Asmi, E., Aalto, P. P., Petäjä,
T., Kerminen, V.-M., and Kulmala, M.: Charged and total particle formation and growth rates during
EUCAARI 2007 campaign in Hyytiälä, Atmos. Chem. Phys., 9, 4077–4089,
https://doi.org/10.5194/acp-9-4077-2009, 2009.
Manninen, H. E., Nieminen, T., Asmi, E., Gagné, S., Häkkinen, S., Lehtipalo, K., Aalto, P., Vana,
M., Mirme, A., Mirme, S., Hõrrak, U., Plass-Dülmer, C., Stange, G., Kiss, G., Hoffer, A., Törő, N.,
Moerman, M., Henzing, B., de Leeuw, G., Brinkenberg, M., Kouvarakis, G. N., Bougiatioti, A.,
Mihalopoulos, N., O'Dowd, C., Ceburnis, D., Arneth, A., Svenningsson, B., Swietlicki, E., Tarozzi,
L., Decesari, S., Facchini, M. C., Birmili, W., Sonntag, A., Wiedensohler, A., Boulon, J., Sellegri, K.,
Laj, P., Gysel, M., Bukowiecki, N., Weingartner, E., Wehrle, G., Laaksonen, A., Hamed, A.,
Joutsensaari, J., Petäjä, T., Kerminen, V.-M., and Kulmala, M.: EUCAARI ion spectrometer
measurements at 12 European sites – analysis of new particle formation events, Atmos. Chem. Phys.,
10, 7907–7927, https://doi.org/10.5194/acp-10-7907-2010, 2010.
Merikanto, J., Spracklen, D. V., Mann, G. W., Pickering, S. J., and Carslaw, K. S.: Impact of
nucleation on global CCN, Atmos. Chem. Phys., 9, 8601–8616, 2009.
Mirme, S. and Mirme, A.: The mathematical principles and design of the NAIS – a spectrometer for
the measurement of cluster ion and nanometer aerosol size distributions, Atmos. Meas. Tech., 6,
1061–1071, https://doi.org/10.5194/amt-6-1061-2013, 2013.
Paasonen, P., Peltola, M., Kontkanen, J., Junninen, H., Kerminen, V.-M., and Kulmala, M.:
Comprehensive analysis of particle growth rates from nucleation mode to cloud condensation nuclei
in boreal forest, Atmospheric Chemistry and Physics, 18, 12085–12103, https://doi.org/10.5194/acp-
524      18-12085-2018, 2018.

Pichelstorfer, L., Stolzenburg, D., Ortega, J., Karl, T., Kokkola, H., Laakso, A., Lehtinen, K. E. J.,
Smith, J. N., McMurry, P. H., and Winkler, P. M.: Resolving nanoparticle growth mechanisms from
size- and time-dependent growth rate analysis, Atmospheric Chemistry and Physics, 18, 1307–1323,
https://doi.org/10.5194/acp-18-1307-2018, 2018.
Riccobono, F., Rondo, L., Sipilä, M., Barmet, P., Curtius, J., Dommen, J., Ehn, M., Ehrhart, S.,
Kulmala, M., Kürten, A., Mikkilä, J., Paasonen, P., Petäjä, T., Weingartner, E., and Baltensperger,
U.: Contribution of sulfuric acid and oxidized organic compounds to particle formation and growth,
Atmospheric Chemistry and Physics, 12, 9427–9439, https://doi.org/10.5194/acp-12-9427-2012,
533      2012.

Rose, C., Zha, Q., Dada, L., Yan, C., Lehtipalo, K., Junninen, H., Mazon, S. B., Jokinen, T., Sarnela,
N., Sipilä, M., Petäjä, T., Kerminen, V.-M., Bianchi, F., and Kulmala, M.: Observations of biogenic
ion-induced cluster formation in the atmosphere, Science Advances, 4, eaar5218,
https://doi.org/10.1126/sciadv.aar5218, 2018.
Sihto, S.-L., Kulmala, M., Kerminen, V.-M., Dal Maso, M., Petäjä, T., Riipinen, I., Korhonen, H.,
Arnold, F., Janson, R., Boy, M., Laaksonen, A., and Lehtinen, K. E. J.: Atmospheric sulphuric acid
and aerosol formation: implications from atmospheric measurements for nucleation and early growth
mechanisms, Atmospheric Chemistry and Physics, 6, 4079–4091, https://doi.org/10.5194/acp-6-
542      4079-2006, 2006.

Stolzenburg, D., Fischer, L., Vogel, A. L., Heinritzi, M., Schervish, M., Simon, M., Wagner, A. C.,
Dada, L., Ahonen, L. R., Amorim, A., Baccarini, A., Bauer, P. S., Baumgartner, B., Bergen, A.,
Bianchi, F., Breitenlechner, M., Brilke, S., Mazon, S. B., Chen, D., Dias, A., Draper, D. C., Duplissy,
J., Haddad, I. E., Finkenzeller, H., Frege, C., Fuchs, C., Garmash, O., Gordon, H., He, X., Helm, J.,
Hofbauer, V., Hoyle, C. R., Kim, C., Kirkby, J., Kontkanen, J., Kürten, A., Lampilahti, J., Lawler,
M., Lehtipalo, K., Leiminger, M., Mai, H., Mathot, S., Mentler, B., Molteni, U., Nie, W., Nieminen,
T., Nowak, J. B., Ojdanic, A., Onnela, A., Passananti, M., Petäjä, T., Quéléver, L. L. J., Rissanen, M.
P., Sarnela, N., Schallhart, S., Tauber, C., Tomé, A., Wagner, R., Wang, M., Weitz, L., Wimmer, D.,
Xiao, M., Yan, C., Ye, P., Zha, Q., Baltensperger, U., Curtius, J., Dommen, J., Flagan, R. C.,
Kulmala, M., Smith, J. N., Worsnop, D. R., Hansel, A., Donahue, N. M., and Winkler, P. M.: Rapid
growth of organic aerosol nanoparticles over a wide tropospheric temperature range, PNAS, 115,
9122–9127, https://doi.org/10.1073/pnas.1807604115, 2018.
Stolzenburg, D., Cai, R., Blichner, S. M., Kontkanen, J., Zhou, P., Makkonen, R., Kerminen, V.-M.,
Kulmala, M., Riipinen, I., and Kangasluoma, J.: Atmospheric nanoparticle growth, Rev. Mod. Phys.,
95, 045002, https://doi.org/10.1103/RevModPhys.95.045002, 2023.
Tuovinen, S., Lampilahti, J., Kerminen, V.-M., and Kulmala, M.: Intermediate ions as indicator for
local new particle formation, Aerosol Research, 2, 93–105, https://doi.org/10.5194/ar-2-93-2024,
560    2024.

Yli-Juuti, T., Nieminen, T., Hirsikko, A., Aalto, P. P., Asmi, E., Hõrrak, U., Manninen, H. E.,
Patokoski, J., Dal Maso, M., Petäjä, T., Rinne, J., Kulmala, M., and Riipinen, I.: Growth rates of
nucleation mode particles in Hyytiälä during 2003-2009: variation with particle size, season, data
analysis method and ambient conditions, Atmos. Chem. Phys., 11, 12865–12886,
https://doi.org/10.5194/acp-11-12865-2011, 2011.
Yu, F. and Luo, G.: Simulation of particle size distribution with a global aerosol model: contribution
of nucleation to aerosol and CCN number concentrations, Atmos. Chem. Phys., 9, 7691–7710, 2009.
Zhao, B., Donahue, N. M., Zhang, K., Mao, L., Shrivastava, M., Ma, P.-L., Shen, J., Wang, S., Sun,
J., Gordon, H., Tang, S., Fast, J., Wang, M., Gao, Y., Yan, C., Singh, B., Li, Z., Huang, L., Lou, S.,
Lin, G., Wang, H., Jiang, J., Ding, A., Nie, W., Qi, X., Chi, X., and Wang, L.: Global variability in
atmospheric new particle formation mechanisms, Nature, 631, 98–105,
https://doi.org/10.1038/s41586-024-07547-1, 2024.