# Peer review of "A Cross-Correlation Based Method for Determining"

_Aerosol Research, 2025_

## Author Comment (AC1)

**Response to Comments by Anonymous Referee #1**

This manuscript by Lampilahti et al. presents a new method to calculate the particle growth rate (GR) during new particle formation (NPF). The particle GR is a key metric to understand the gas-to-particle conversion during NPF and the calculation from the particle number size distribution (PNSD) is still a considerable source of uncertainty when mechanistic insights into NPF are inferred from ambient aerosol data. This has two reasons: 1) considerable instrumental uncertainties in the sub-10 nm size range 2) the usage of GR calculation methods which highly depend on the user or are sensitive to the measurement uncertainties. In that sense, this manuscript presents a considerable step forward for the community working on analyzing NPF as no fully automated method to infer the GR from the PNSD has been presented so far. The introduced maximum cross-calculation method is shown to be robust compared to the other methods when a comparing the results for Hyytiälä Finland. However, the manuscript in its current form has some shortcomings, which should be addressed such that this paper encourages the involved community to use that new method in the future and therefore I can only recommend publication in AR after the following points have been addressed:

We thank the reviewer for the comments. Please see our responses below in blue text color.

Major comments:

The authors show in this manuscript that the new maximum cross-correlation method performs similar (providing the same medians and variance) to other methods when applied to a dataset from Hyytiälä, Finland. What is missing from this manuscript (and what should be part of an introduction into any new method) is the testing of its performance against data where the true GR is known! The central point of this manuscript is not only to show that the new approach reproduces probably the same errors than the other methods have but to show that it probably can also retrieve the actual GR of an NPF event. Inclusion of a synthesized dataset (with instrumental uncertainties imposed) and the application of the new method to it should be the number 1 point of the results. I am 100% confident,

that this group of authors has access to such synthesized NPF event data and it is therefore not too much additional work to include.

The second major point relates to the fact that the authors analyze 14 years of Hyytiälä data, but then do not make use of the already existing analyses of these datasets. As far as I know, already analyzed Hyytiälä NPF event classification and GR data should be available to that set of authors for the same 14 years they now present as being analyzed by the maximum cross-correlation method. It is unclear to my why Figure 5, only contains randomly selected days where the GR is re-analyzed with the maximum concentration method, while there should be the entire GR dataset for 14 years be available. There are two specific things I'd like to see in a revised manuscript: 1) Correlation plots of the full 14-year dataset for all instruments (ion-GR, particle-GR NAIS, particle-GR DMPS) between all GRs calculated from the maximum cross-correlation method and GRs calculated from other methods, whenever both methods returned a value. 2) The background/signal differentiation histograms when only data from manually classified NPF events (or use the ranking method or whatever) are used. Does the background population completely vanish if we only use values from "classified" NPF events (should be Appendix Figure).

We made major revisions to the results section with a focus on validating the method.

1) We investigated the sensitivity of the method in response to varying input parameters (normalization*, number of bins, maximum allowed time lag, smoothing window width) (Figures 2 and 3).
2) We simulated a NPF event with known GR and compared the performance of our method and the conventional methods (Figure 4)
3) We compared our results from Hyytiälä against a pre-existing GR dataset (2010-2019) (Figures 7 and 8)
4) We included a low nanorank vs high nanorank comparison to the GR distribution plot (Figure 5). At high nanorank values the background was reduced.
5) We compared the days with valid GRs against NPF event classification and nanoranking (Figure 9)

From the previous version Figures 5, 6 and 7 were now not included as they did not contribute so much to the method's validation.

*We added another input parameter (gamma) which controls the degree of overlap correction when the particle concentration time series are cross-correlated. We selected the gamma based on the condition that the GR should not change if the size range is divided into smaller and smaller bins.

Minor comments:

Line 35-42: Stolzenburg et al., 2023, Rev. Mod. Phsy. is the most recent and complete review on particle growth and also compares different methods with each other. It should be included here.

We included the reference. We also found the terminology used in the article useful (e.g. representative diameter, representative time) so we adopted some of the terms in our manuscript.

Line 37: In the above mentioned review, there is a long discussion about a third approach using the full evolution of the PNSD combined with the general dynamics equations. These methods do disentangle the different contributions to GR but suffer from other challenges. They should be mentioned here, because especially those methods would have the chance to also be fully automated (in an ideal world).

We added a discussion on dynamic methods to the introduction.

Line 63: What is the difference between these methods to the one presented in Lehtipalo et al., 2014. This should be clarified here.

We added the following clarification: "The difference between finding representative time lags and representative times is subtle but relevant since in practice the methodology can be different and one may be more amenable to automation than the other."

Line 67: I would also add that one of the advantages of the size channel based methods is that they do not require perfect knowledge on the absolute concentrations of the PNSD (i.e. the inversion correctness). In fact, they sometimes can even be run on raw data. The same applies to the new maximum cross-correlation method and should be mentioned here.

We added this to the advantages

Line 100-101: I'd prefer $N\_1$ (with bar above) for the mean notation, as this is far more common.

There seems to be a bug in the pdf conversion, the bar should be above. We changed the symbol to Greek letter mu to avoid trouble.

Line 109: To judge the robustness of the method, it would also be interesting how the results change when a different averaging is applied. Especially if the method is transferred to other environments, this parameter might need to change. In addition, it is known that rolling averages can skew the results of the appearance time method, so it would be interesting to see what changes when a static, time resolution reduction (i.e. block averages) is used.

Sensitivity test to the smoothing window width was added (see summary of major revisions). Adding more smoothing does skew the results towards lower GRs.

Line 123-131: It is not fully clear to me how the approach in creating more size increments is facilitated. Is the PNSD first inverted for all data and then somehow resized? I.e. how many original channels are e.g. in that window between 2-3 nm and how many increments are later used? I.e. do you obtain a higher size-resolution than the original data with that approach?

In our version of the algorithm the size range is divided into n equally sized bins on log scale. The concentrations are interpolated in linear piecewise manner to the diameters that divide the bins.

Having more bins than there are size channels in the size range will result in tau_max=0 evaluations and no GR output. We added to the Methods section 2.1 in the revised version that one should not exceed the number of size channels when dividing into bins. The log distance between size channels in the NAIS data is at most 0.03 and in the DMPS data 0.05. So between 2-3 nm there are 6 size channels (we divide it into 2 bins).

Line 133: In my opinion the method should be called maximum cross-correlation method throughout the manuscript as maximum correlation method could be misleading/ doesn't describe exactly what the method does.

We changed the naming and abbreviated it as MCC in the revised text.

Line 147-149: That sentence seems to be broken. "To ensure" what?

Removed "to ensure"

Line 149-151: Again, how many original size channels are in these ranges?

Number of channels between 2-3 nm in NAIS: 6
Number of channels between 3-7 nm in NAIS: 12
Number of channels between 7-20 nm in NAIS: 15
Number of channels between 3-7 nm in DMPS: 7
Number of channels between 7-20 nm in DMPS: 9

We added this information to the revised manuscript section 2.2 describing Hyytiälä dataset

Line 173-178 and 183-184: As said in the major comment: Here should be a comparison to the classical NPF event day characterization.

See the summary of major revisions above

Line 192-194: Probably the right argument. But assuming these fast GRs are really there, would it make sense to then probably just tolerate e.g. one of the tau_max,i to be below zero to still capture some of these?

With the new normalization using the exponent gamma=0.25 the shift in the GR distribution is less pronounced. Some of the shift is from bias towards smaller GRs caused by the previous normalization where gamma=1 (see revised Figure 2). The bias accumulates when the size range is divided into more bins. We tried accepting some invalid tau_max,i and it did indeed bring some extra days, but the difference was so small that in our opinion it did not justify making it the default option.

Line 211-214: As the new method can calculate GR on more days, it would be very interesting to see how this corresponds to a event classification scheme (major comment 2)

See above

Line 216-218, Figure 5: It is important to see this comparison across different instruments and also probably different methods. Moreover, as said in major comment 2, the authors should make use of the full Hyytiälä datasets available to them.

See outline of the major revisions above. The instruments in the pre-existing GR dataset matched the ones we used (NAIS ion mode and DMPS). The method in the pre-existing dataset was maximum concentration, unfortunately no other methods were used.

Line 240-241: Again Stolzenburg et al. (2023) probably provides the to-date most complete overview of GR datasets and should be referenced here.

Citation added, see below

Line 245-250: Feels a bit off here, as this more or less is Figure 8, but Figure 6 is not yet discussed here. Should that be later in the manuscript?

The discussion on median GRs and their size dependent behavior is now in text where we discuss Figure 5. We only describe the NAIS total particles shortly. This is the modified paragraph:

"The median GRs in Figure 5 showed an increasing trend with particle size, a pattern commonly observed across various environments (Kerminen et al., 2018; Stolzenburg et al., 2023). In the same size range the median GRs found for the positive ions, negative ions and total particles (from DMPS) were similar, which is also supported by previous observations (Gonzalez Carracedo et al., 2022; Hirsikko et al., 2005; Manninen et al., 2009; Yli-Juuti et al., 2011). The GRs calculated from the NAIS total particles were higher compared to the other GRs in the same size range. This might be linked to the NAIS measuring higher concentrations in total particle mode compared to other instruments (Kangasluoma et al., 2020)."

Line 253-255, Figure 6: My honest opinion: This Figure is not very interesting as it doesn't provide any new insights (except that the new method can perform across different seasons). If the authors want to save space as they need to include the comparisons with synthesized data or the full Hyytiälä dataset, they can remove it. In addition, the caption is too short and should explain more what is in that Figure.

This figure was removed

Line 260-268, Figure 7: I would love to see the median days also for the "edges" of the GR distribution, i.e. the very fast and very slow growth cases, as these might be the most interesting, where deviations from the classical "banana-type" picture might appear.

We found this figure to be too much on the side of interesting results than method validation. Therefore to keep the manuscript coherent we decided to leave it out.

Line 296-297: Why only ion GR from Gonzalez-Carracedo? The DMA train data are especially useful in the sub 3 nm range, where other instruments often perform worse. This comparison should be shown, as this is one of the cases where the method might reach its limits (I don't think so, but it needs to be included).

We did not calculate the GR from the DMA train data using our method in the sub-3 nm size range. We only used NAIS ions, therefore in order to compare "apples to apples" we only included the NAIS ion data from Gonzalez-Carracedo in the comparison.

---

## Author Comment (AC2)

**Response to Comments by Anonymous Referee #2**

The work of Lampilahti et al. introduces a novel, fully automatic method for calculating particle growth rates (GRs) during new particle formation (NPF) events, using a maximum correlation approach applied to particle number size distribution data (both ions and particles). The accurate determination of GRs is essential for understanding NPF mechanisms and for quantifying the influence of NPF on climate, particularly in relation to the formation of CCN. Automating this calculation is a crucial, but challenging step and I consider the current manuscript a valuable addition to the NPF community. However, in its current form, the manuscript presents a few methodological and conceptual issues that need to be addressed to ensure the robustness, transparency, and comparability of the proposed method. I therefore recommend the manuscript for publication in AR after the authors have adequately addressed the major and minor comments outlined below.

We thank the reviewer for the comments, see our responses below in blue text color.

Major comment:

The GRs calculated with the maximum correlation method are compared with GRs calculated with other methods, for the corresponding time periods, in Hyytiälä. However, the various studies used for comparison have utilized an event classification algorithm (either manual or automatic) before calculating the GRs. These are usually calculated only for clear/strong (Class I) NPF events but sometimes even for Class II events (Manninen et al., 2009). The new method seems to include days not typically classified as clear events (maybe sometimes undefined but also non-events due to quiet NPF). While I agree that most "Signal Days" are indeed clear NPF events, I have concerns about the comparability of the results, because of lack of classification prior to GR calculation. The absence of a pre-classification step introduces uncertainty; for example, your method may exclude days previously identified as Class I events or include days that would not typically qualify. What was the mean clear event (Class I) frequency in Hyytiälä during the periods of interest? Is it comparable with the signal % you find in Fig. 4? How many days were previously classified as Class I but were excluded by your method (and vice versa)?

One can certainly classify or filter the data prior to (or after) calculating the GRs. We applied the method to all days since particle growth is not restricted to only intense NPF event days. In the revised manuscript we compared the maximum cross-correlation (MCC) method against the maximum concentration (MC) method only on overlapping days between 2010-2019 (see the revised Figure 7) so classification should not be a factor in that comparison.

In the revised manuscript we also prepared Figure 9 to study how the GR days and non GR days from the MCC method were distributed to traditional NPF event classes. The fraction of Class I NPF events of all days was about 10% while for example GR_3-7(-) was calculated about 12% of the days (Figure 6b). Figure 9 shows that about 40% of GR_3-7(-) days were Class I NPF event days, 30% were Class II NPF events, 20% were undefined days and 10% were nonevent days. Therefore around half of the Class I NPF events get a GR_3-7(-) by our method.

By comparison in the manually calculated ion GR dataset we used in the revised manuscript GR_3-7(-) was calculated on 4.5% of the days (see revised Figure 8). Assuming all these were Class I NPF events, then 45% of all Class I days were assigned a GR_3-7(-), which is similar to the fraction seen in our method.

The point is that there can be large variations in the number of days when GR was successfully calculated depending on the dataset (using conventional method). Similar subjectivity exists in the NPF event classification itself and one should keep this in mind when looking at the number of NPF events. What one can say is that most days when our method found particle growth beyond 3 nm were NPF event days.

Furthermore, as pointed out in lines 167-172 (and elsewhere) the early morning/late evening concentrations leading to erroneous and small GRs (background) seems to be a small weakness of the method (especially if one wants to use it in environments with increased local sources etc.), which is of course expected for automatic methods. However, my greatest concern is the possibility of getting GRs that appear reasonable, but for the wrong reasons, for example by having similar concentration and diameter increases from different sources (and not NPF) which can "trick" the method into calculating a "fake" GR higher than β, resulting in "signal" and not in "background". Figure 7 helps illustrate that most of these days are NPF, but because it is averaged and normalized, days that do not follow this behavior can be averaged out. Have you observed such discrepancies in Hyytiälä?

Since the method is quite simple there are certainly cases where non-NPF processes contributed to the "signal" in Hyytiälä. However, as was discussed above, most GR days from the signal part are NPF event days.

General comment:

I suggest that the "Results" section should be reorganized into clearly defined subsections. At present, the narrative lacks coherence, and transitions between paragraphs are abrupt, making the analysis difficult to follow. For instance, the paragraph starting on line 260 appears disconnected from the preceding discussion and could logically belong to a separate subsection

We have done major revisions to the results section focusing on validation of the method. We hope the text is easier to follow after these changes.

Minor comments:

Lines 54-67: I would suggest commenting a bit more about how these existing methods compare with each other. When each method is preferred (for example type of data, available sizes etc.).

The conditions when to use each type of method were in the text but we clarified the points further in the revised version.

Line 63: A related approach to what? Related to the size channel-based methods? Please clarify.

Replaced with word "similar"

Lines 70-72: Please elaborate a little bit more. GR calculation and especially fitting-based methods are extremely sensitive to concentration spikes (e.g. from local sources), noisy data and even meteorological conditions that can alter the number concentration for a while.

We elaborated on this point.

Lines 73-74: Additionally, there is not a "universal" GR calculation method, and some (older) studies do not even mention how GR was calculated exactly.

We agree and added that the methods used can be poorly documented.

Lines 96-97: What was the temporal resolution of your dataset? It would help to specify it here.

We mentioned it in the next subsection when describing the dataset (1 hour resolution)

Lines 109-110: Isn't that a high averaging time for such a dynamic phenomenon as the growth rate? I get that in Hyytiälä the GRs are generally small compared to other environments, but still when calculating it for the size range of 2-3 nm (or even the 3-7 nm) as you do later, major changes can happen even within 1 hour. Also, I'm certain that during the 14 years, there should be NPF days with high GRs. Is the method/ averaging/ smoothing time-sensitive enough to catch these dynamic changes and calculate a trustworthy GR? What happens if you have more than one NPF events on the same day?

Probably very fast growth is lost due to temporal averaging but the regional NPF phenomena should be detected. If there are more than one NPF event during one day the GR would probably be close to a concentration weighted average GR for the two NPF events. The method can be run with higher temporal resolution data and the time window can be changed from 1 day to something else (the time windows can be asymmetric in length) if the growth process is fast.

Figure 1: What is the data time resolution you use in general with this method? From Fig 1a it seems it's 1 hour. However, you have said that you use 3 hour rolling mean smoothing to run the method. Why not apply it in these graphs?

We added the normalized number concentrations before and after smoothing to the figure. In general one can use any time resolution with the method. With regional NPF events the ideal range is probably somewhere between 30 min-1 hour.

Line 167-168: How was the local minimum, β ranging between the different diameter classes? In other words, what was the minimum growth rate above which you considered to have an NPF event (signal)?

We marked the cutoffs to the plots.

Line 171-172: I understand that, but it is not very clear to me why this happens. Please elaborate on this.

We added more explanation in the revised version.

Line 177: Please replace "of" with "or".

Replaced "of" with "or"

Line 181: How many days were the outliers? What was different in these days? Increased local sources?

We added to the text that it was less than 20 in all GR distributions. The reason would most likely be something that causes a sudden change in the number size distribution such as local emission or air mass change.

Fig A1 and A3 and A5 (and also lines 175-178): It is related to my major comment. You say that "the background days mostly days that would be classified as non-event or undefined days". I agree, but also the signal days contain days that would not be necessarily classified as "Class I" days (which are the class typically used for GR calculations in most studies). In fact, from Figs A1, A3, A5 I see days that I would classify as non-events (Fig. A1 middle, and second line to the left) or undefined (Fig. A3 in the middle, and the one bottom right) (and also some are close to Class II). Even if a growth pattern exists in these cases (which is not clear in this color scale) how do these GRs compare to the actual NPF observations of the previous studies that use only Class I (and sometimes also Class II) events to calculate the GR?

We prepared Figure 7 in the revised manuscript which shows how well the new method correlates with conventionally calculated GRs in Hyytiälä (2010-2019).

Lines 189-196: How did you decide the 0.1 log difference for the size increments? Was it a trial-error approach? With this approach, how many high-GR days do you think were discarded because of $\tau_{max}=0$? Could lowering the data averaging time help for these days?

Trial and error approach, it seemed a good compromise in signal-to-noise ratio vs amount of days returning GR. Using better time resolution might help to return GR on days when the growth is very fast.

Lines 212-214: I suspect that the higher percentages of your method are mostly because of the so-called "quiet NPF" (Kulmala et al., 2022). Since you normalize the number concentrations, the algorithm detects growth patterns not detected by previous manual methods. Is that correct? Can you elaborate on that? If this is true, this can actually be a good feature about this method (meaning to calculate GR of days that would be otherwise not included manually).

This is right

Lines 215-219: As Reviewer #1 pointed out, I would like to see a more comprehensive comparison of the new method with previous calculations of the GRs in Hyytiälä. Maybe you could utilize a GR database and include more days in Fig. 5.

We added much more comparison to the revised manuscript.

Lines 245-246 and 249-250: Why is that? Please elaborate.

We are not sure why the NAIS total particles return higher GRs. We could not find any data to compare against. We added some speculation though:

> "This might be linked to the NAIS measuring higher concentrations in total particle mode compared to other instruments (Kangasluoma et al., 2020)."

Figure 6: This Figure is not adequately described in both the main text and the figure's caption. Furthermore, it is not connected to the previous analyses of the paper, and I believe it is of small value in the current paper (at least in the main text) since it focuses on the GR calculation method and not its general variability in the atmosphere of Hyytiälä. Maybe you could move it to the Supplement or try to connect it better to the rest of the text.

We removed Figure 6 and the related parts in the text as it is of less relevance.

Lines 260-268: I'm not sure what the conclusion of this paragraph is. That the days included automatically by the method indeed represent NPF days? Why don't you first use an automatic classification method (Aliaga et al., 2023) to make sure that you have NPF events only? Because Fig. 6 is the median normalized diurnal, the days that are not typical NPF events but are included in the GR calculation are averaged out.

Lines 267-268: Isn't that self-evident?

We removed the Figure and this paragraph. To investigate if GR days are related to NPF we prepared the revised Figure 9.

Lines 275-303: Relates to major comment. The studies you are comparing with have used only the clear NPF events (usually classified as Class I but some use also the Class II) to calculate the GRs, so their averages may include less days than the ones included in the new method. I think it is important to clarify this here and elsewhere in the text. Additionally, in Figure 8, consider annotating each bar with the number of days included in the corresponding average (maybe above the bars).

We added the following clarifying statement:

> "In the previously published studies the GRs were determined only on days with intense NPF, whereas our method is in principle also capable of finding GRs on days with less intense NPF. In Hyytiälä the GR is expected to be similar despite the intensity of NPF (Kulmala et al., 2022)."

The number of days the GR was calculated in each size range was unclear in the previous studies, so we did not add it to the figure as annotation.

Lines 311-312: This statement can be misleading, since the method was tested and validated only for Hyytiälä and no other environments yet. Please rephrase.

We rephrased it to: "The method was tested and validated on sub-20 nm ion and total particle number size distribution data measured from a Finnish boreal forest."

Lines 339-342: That is a good idea, to combine the method with the automatic classification. From my point of view, the classification of the events should always come before the GR calculation, which is something the authors do not do directly. Have the authors tried this combination? It could increase the efficiency and the comparability of the method, since days not typically classified as NPF will not be included in the GR calculation.

This is something to be done in the future. In revised Figure 5 we plotted the GR distribution on days with high nanorank and low nanorank to show the reduction in "background" in high nanorank days.

In a way our method decouples the GR calculation step from the classification step, since no matter how you classify the data the same days will always have the same GR. One can use the whole dataset to explore correlations between different variables. If classification/filtering is wanted it can also be done but is not required in order to get the GRs.

As discussed in Kulmala et al. (2024) including all days in the NPF analysis is likely to be a fruitful approach in the future.

**References**

Kulmala, M., Aliaga, D., Tuovinen, S., Cai, R., Junninen, H., Yan, C., Bianchi, F., Cheng, Y., Ding, A., Worsnop, D. R., Petäjä, T., Lehtipalo, K., Paasonen, P., and Kerminen, V.-M.: Opinion: A

paradigm shift in investigating the general characteristics of atmospheric new particle formation using field observations, Aerosol Research, 2, 49–58, https://doi.org/10.5194/ar-2-49-2024, 2024.

References

Aliaga, D., Tuovinen, S., Zhang, T., Lampilahti, J., Li, X., Ahonen, L., Kokkonen, T., Nieminen, T., Hakala, S., Paasonen, P., Bianchi, F., Worsnop, D., Kerminen, V.-M., and Kulmala, M.: Nanoparticle ranking analysis: determining new particle formation (NPF) event occurrence and intensity based on the concentration spectrum of formed (sub-5 nm) particles, Aerosol Research, 1, 81–92, https://doi.org/10.5194/ar-1-81-2023, 2023.

Kulmala, M., Junninen, H., Dada, L., Salma, I., Weidinger, T., Thén, W., Vörösmarty, M., Komsaare, K., Stolzenburg, D., Cai, R., Yan, C., Li, X., Deng, C., Jiang, J., Petäjä, T., Nieminen, T., and Kerminen, V. M.: Quiet New Particle Formation in the Atmosphere, Frontiers in Environmental Science, 10, https://doi.org/10.3389/fenvs.2022.912385, 2022.

Manninen, H. E., Nieminen, T., Riipinen, I., Yli-Juuti, T., Gagné, S., Asmi, E., Aalto, P. P., Petäjä, T., Kerminen, V.-M., and Kulmala, M.: Charged and total particle formation and growth rates during EUCAARI 2007 campaign in Hyytiälä, Atmospheric Chemistry and Physics, 9, 4077–4089, https://doi.org/10.5194/acp-9-4077-2009, 2009.

Citation: https://doi.org/10.5194/ar-2025-19-RC2